# LAVAE: Disentangling Location and Appearance

## Abstract

We propose a probabilistic generative model for unsupervised learning of structured, interpretable, object-based representations of visual scenes. We use amortized variational inference to train the generative model end-to-end. The learned representations of object location and appearance are fully disentangled, and objects are represented independently of each other in the latent space. Unlike previous approaches that disentangle location and appearance, ours generalizes seamlessly to scenes with many more objects than encountered in the training regime. We evaluate the proposed model on multi-MNIST and multi-dSprites data sets.

## 1 Introduction

Many hallmarks of human intelligence rely on the capability to perceive the world as a layout of distinct physical objects that endure through time—a skill that infants acquire in early childhood (Spelke, 1990; 2013; Spelke and Kinzler, 2007). Learning compositional, object-based representations of visual scenes, however, is still regarded as an open challenge for artificial systems (Bengio et al., 2013; Garnelo and Shanahan, 2019).

Recently, there has been a growing interest in unsupervised learning of disentangled representations (Locatello et al., 2018), which should separate the distinct, informative factors of variations in the data, and contain all the information on the data in a compact, interpretable structure (Bengio et al., 2013). This notion is highly relevant in the context of visual scene representation learning, where distinct objects should arguably be represented in a disentangled fashion. However, despite recent breakthroughs (Chen et al., 2016; Higgins et al., 2017; Kim and Mnih, 2018), multi-object scenarios are rarely considered (Eslami et al., 2016; Van Steenkiste et al., 2018; Burgess et al., 2019).

We propose the Location-Appearance Variational AutoEncoder (LAVAE), a probabilistic generative model that, without supervision, learns structured, compositional, object-based representations of visual scenes. We explicitly model an object's location and appearance with distinct latent variables, unlike in most previous works, thus providing a highly beneficial inductive bias. Following the framework of variational autoencoders (VAEs) (Kingma and Welling, 2013; Rezende et al., 2014), we parameterize the approximate variational posterior of the latent variables with inference networks that are trained end-to-end with the generative model. Our model learns to correctly count objects and compute a compositional, object-wise, interpretable representation of the scene. Objects are represented independently of each other, and each object's location and appearance are disentangled. Unlike previous approaches that disentangle location and appearance, LAVAE generalizes seamlessly to scenes with many more objects than in the training regime. We demonstrate these capabilities on multi-MNIST and multi-dSprites data sets similar to those by Eslami et al. (2016) and Greff et al. (2019).

## 2 Method

**Generative model.** We propose a latent variable model for images in which the latent space is factored into location and appearance of a variable number of objects. For each image $\mathbf{x}$ with $D$ pixels, the number of objects is modeled by a latent variable $n$, their locations by $n$ latent variables $\{\mathbf{z}_{\mathrm{loc}}^{(i)}\}_{i=1}^n$, and their appearance by $\{\mathbf{z}_{\mathrm{app}}^{(i)}\}_{i=1}^n$. We assume the number of objects in every image to

be bounded by $M$. The joint distribution of the observed and latent variables for each data point is:

$$p(\mathbf{x}, \mathbf{z}_{\text{loc}}, \mathbf{z}_{\text{app}}, n) = p(\mathbf{x} \,|\, \mathbf{z}_{\text{loc}}, \mathbf{z}_{\text{app}}) p(\mathbf{z}_{\text{loc}} \,|\, n) p(\mathbf{z}_{\text{app}} \,|\, n) p(n)$$

$$= p(\mathbf{x} \,|\, \mathbf{z}_{\text{loc}}, \mathbf{z}_{\text{app}}) \left( \prod_{i=1}^{n} p\Big(\mathbf{z}_{\text{loc}}^{(i)} \,|\, \{\mathbf{z}_{\text{loc}}^{(j)}\}_{j<i}\Big) p(\mathbf{z}_{\text{app}}^{(i)}) \right) p(n) , \tag{1}$$

where we use the shorthand $\mathbf{z}_{\text{app}} = \{\mathbf{z}_{\text{app}}^{(i)}\}_{i=1}^{n}$ and similarly for $\mathbf{z}_{\text{loc}}$.

The generative process can be described as follows. First, the number of objects $n$ is sampled from a categorical distribution

$$p(n) = \text{Cat}(M + 1, \mathbf{a}), \qquad n \in \{0, \ldots, M\} , \tag{2}$$

where $\mathbf{a}$ is a learned probability vector of size $M + 1$. The $n$ location variables are sequentially sampled without replacement from a categorical distribution with $D$ classes:

$$p\Big(\mathbf{z}_{\text{loc}}^{(i)} \,|\, \{\mathbf{z}_{\text{loc}}^{(j)}\}_{j<i}\Big) = \text{Cat}(D, \mathbf{b}^{(i)}), \qquad \mathbf{b}^{(i)} = \frac{1}{D - i + 1} \left( \mathbf{1}_D - \sum_{j=1}^{i-1} \mathbf{z}_{\text{loc}}^{(j)} \right) , \tag{3}$$

where $i = 1, \ldots, n$, and $\mathbf{1}_D$ is a vector of ones of length $D$. To each $\mathbf{z}_{\text{loc}}^{(i)}$, which is a one-hot representation of an object's location, corresponds a continuous appearance vector $\mathbf{z}_{\text{app}}^{(i)} \sim \mathcal{N}(\mathbf{0}_L, \mathbf{I}_L)$ of size $L$ that describes the object.

The likelihood function $p(\mathbf{x} \,|\, \mathbf{z}_{\text{loc}}, \mathbf{z}_{\text{app}})$ is parameterized in a compositional manner. For each image, the visual representation, or *sprite*, of the $i$th object is generated from $\mathbf{z}_{\text{app}}^{(i)}$ by a shared function $f_{\text{sprite}}$. Each sprite is then convolved with a 2-dimensional Kronecker delta that is the one-hot representation of the object's location. Finally, the resulting $n$ tensors are added together to give the pixel-wise parameters $\boldsymbol{\lambda} = (\lambda_1, \ldots, \lambda_D)$ of the distribution:

$$\boldsymbol{\lambda} = \sum_{i=1}^{n} f_{\text{sprite}}\big(\mathbf{z}_{\text{app}}^{(i)}\big) * \mathbf{z}_{\text{loc}}^{(i)} , \tag{4}$$

where $*$ denotes 2-dimensional discrete convolution.

**Inference model.** The approximate posterior for a data point $\mathbf{x}$ has the following form:

$$q(\mathbf{z}_{\text{app}}, \mathbf{z}_{\text{loc}}, n \,|\, \mathbf{x}) = q(\mathbf{z}_{\text{app}} \,|\, \mathbf{z}_{\text{loc}}, n, \mathbf{x}) q(\mathbf{z}_{\text{loc}} \,|\, n, \mathbf{x}) q(n \,|\, \mathbf{x})$$

$$= \left( \prod_{i=1}^{n} q(\mathbf{z}_{\text{app}}^{(i)} \,|\, \mathbf{z}_{\text{loc}}^{(i)}, \mathbf{x}) q\Big(\mathbf{z}_{\text{loc}}^{(i)} \,|\, \{\mathbf{z}_{\text{loc}}^{(j)}\}_{j<i}, \mathbf{x}\Big) \right) q(n \,|\, \mathbf{x}) . \tag{5}$$

Since each data point is assumed independent, the lower bound (ELBO) to the marginal log likelihood is the sum of the ELBO for each data point, which is:

$$\log p(\mathbf{x}) \geq \mathbb{E}_q \left[ \log p(\mathbf{x} \,|\, \mathbf{z}_{\text{loc}}, \mathbf{z}_{\text{app}}) \right] + \mathbb{E}_q \left[ \log \frac{p(\mathbf{z}_{\text{loc}}, \mathbf{z}_{\text{app}}, n)}{q(\mathbf{z}_{\text{loc}}, \mathbf{z}_{\text{app}}, n \,|\, \mathbf{x})} \right] , \tag{6}$$

where the first term is the likelihood and the second is the negative Kullback-Leibler (KL) divergence between $q(\mathbf{z}_{\text{loc}}, \mathbf{z}_{\text{app}}, n \,|\, \mathbf{x})$ and $p(\mathbf{z}_{\text{loc}}, \mathbf{z}_{\text{app}}, n)$. The different terms of the KL divergence can be derived as in Appendix A and estimated by Monte Carlo sampling.

Two inference networks compute appearance and location feature maps, $\mathbf{h}_{\text{app}}$ and $\mathbf{h}_{\text{loc}}$, both having size $D$ like the input. The inference model for the number of objects $n$ is a categorical distribution parameterized by a function of the location features $f_{\text{count}}(\mathbf{h}_{\text{loc}})$. Object locations follow $n$ categorical distributions without replacement parameterized by logits $\mathbf{h}_{\text{loc}}$. The vector at location $\mathbf{z}_{\text{loc}}^{(i)}$ in the feature map $\mathbf{h}_{\text{app}}$ represents the appearance parameters for object $i$, i.e. the mean and log variance of $q(\mathbf{z}_{\text{app}}^{(i)} \,|\, \mathbf{z}_{\text{loc}}^{(i)}, \mathbf{x})$. The overall inference process can be summarized as follows:

$$q(n \,|\, \mathbf{x}) = \text{Cat}(n; \, M + 1, f_{\text{count}}(\mathbf{h}_{\text{loc}})) \tag{7}$$

$$q\Big(\mathbf{z}_{\text{loc}}^{(i)} \,\Big|\, \{\mathbf{z}_{\text{loc}}^{(j)}\}_{j<i}, \mathbf{x}\Big) = \text{Cat}(\mathbf{z}_{\text{loc}}^{(i)}; \, D, \mathbf{b}^{(i)}) \qquad i = 1, \ldots, n \tag{8}$$

$$\boldsymbol{\mu}_{\mathrm{app}}^{(i)}, \boldsymbol{\sigma}_{\mathrm{app}}^{2\,(i)} = \mathbf{h}_{\mathrm{app}}[\mathbf{z}_{\mathrm{loc}}^{(i)}] \qquad\qquad i = 1, \ldots, n \qquad\qquad (9)$$

$$q(\mathbf{z}_{\mathrm{app}}^{(i)} \mid \mathbf{z}_{\mathrm{loc}}^{(i)}, \mathbf{x}) = \mathcal{N}(\mathbf{z}_{\mathrm{app}}^{(i)}; \boldsymbol{\mu}_{\mathrm{app}}^{(i)}, \mathrm{diag}(\boldsymbol{\sigma}_{\mathrm{app}}^{2\,(i)})) \qquad i = 1, \ldots, n \,, \qquad (10)$$

where by $\mathbf{v}[\mathbf{e}_k]$ we denote the $k$th element of $\mathbf{v}$ (with $\mathbf{e}_k$ a standard basis vector), and the probability vector for location sampling at each step is computed iteratively:

$$\mathbf{b}_0^{(i)} = \mathrm{softmax}(\mathbf{h}_{\mathrm{loc}}) \cdot \left(\mathbf{1}_D - \sum\nolimits_{j=1}^{i-1} \mathbf{z}_{\mathrm{loc}}^{(j)}\right), \qquad \mathbf{b}^{(i)} = \mathbf{b}_0^{(i)} \cdot ||\mathbf{b}_0^{(i)}||_1^{-1} \,. \qquad (11)$$

The expectations in the variational bound are handled as follows: For $n$ we use discrete categorical sampling. This gives a biased gradient estimator, but in practice this has not affected inference on $n$. We use the Gumbel-softmax relaxation (Jang et al., 2016; Maddison et al., 2016) for $\mathbf{z}_{\mathrm{loc}}$ and the Gaussian reparameterization trick for $\mathbf{z}_{\mathrm{app}}$.

## 3 RESULTS

In all experiments we use the same architecture for LAVAE. See Appendix B for all implementation details. Briefly, the sprite decoder in the generative model $p(\mathbf{x} \mid \mathbf{z}_{\mathrm{loc}}, \mathbf{z}_{\mathrm{app}})$ consists of a fully connected layer and a convolutional layer, followed by 3 residual convolutional blocks. The appearance and location inference networks are fully convolutional, thus the feature maps $\mathbf{h}_{\mathrm{app}}$ and $\mathbf{h}_{\mathrm{loc}}$ have the same spatial size as $\mathbf{x}$.

As a baseline to compare the proposed model against we used a VAE implemented as a fully convolutional network: the absence of fully connected layers makes it easier to preserve spatial information, thereby allowing the model to achieve a higher likelihood and generalize more naturally. Moreover, this choice of baseline is closer in spirit to our model than a VAE with fully connected layers.

We evaluate LAVAE on multi-MNIST and multi-dSprites data sets consisting of 200k images with 0 to 3 objects in each image. 190k images are used for training, whereas the remaining 10k are left for evaluation. We generated an additional test set of 10k images with 7 objects each. Examples with more than 3 objects are never used for training.

### 3.1 MULTI-MNIST

Each image in the multi-MNIST data set consists of a number of statically binarized MNIST digits scattered at random (avoiding overlaps) onto a black canvas of size $64 \times 64$. The digits are first rescaled from their original size ($28 \times 28$) to $15 \times 15$ by bilinear interpolation, and finally binarized by rounding. When generating images, digits are picked from a pool of either 1k or 10k MNIST digits. We call the two resulting data sets *multi-MNIST-1k* and *multi-MNIST-10k*, depending on the number of MNIST digits used for generation. We independently model each pixel as a Bernoulli random variable parameterized by the decoder:

$$p(\mathbf{x} \mid \mathbf{z}_{\mathrm{loc}}, \mathbf{z}_{\mathrm{app}}) = \prod_{d=1}^{D} p(x_d \mid \mathbf{z}_{\mathrm{loc}}, \mathbf{z}_{\mathrm{app}}) = \prod_{d=1}^{D} \mathrm{Bernoulli}(x_d; \lambda_d) \,, \qquad (12)$$

where the parameter vector $\boldsymbol{\lambda}$ defined in Section 2 is the pixel-wise mean of the Bernoulli distribution.

Figure 1 qualitatively shows the performance of LAVAE on multi-MNIST-10k test images. The inferred location of all objects in an image are summarized by a black canvas where white pixels indicate object presence. For each of those locations, the model infers a corresponding appearance latent variable. Each digit on the right is generated by $f_{\mathrm{sprite}}$ from one of these appearance variables.

Samples from the prior are shown in Figure 2: note that the number of generated objects is consistent with the training set, since the prior $p(n)$ is learned. Quantitative evaluation results are shown in Table 1 and compared with a VAE baseline. The inferred object count was correct on almost all images of *all test sets*—even with a held-out number of objects—and across all tested random seeds.

A fundamental characteristic of disentangled representations is that a change in a single representational factor should correspond to a change in a single underlying factor of variation (Bengio et al., 2013; Locatello et al., 2018). By demonstrating that the appearance and location of single objects

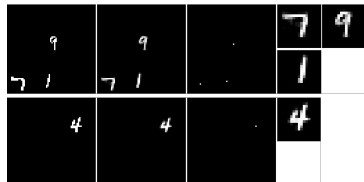 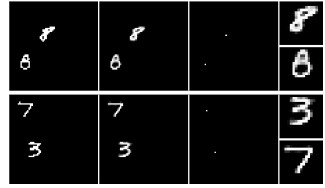

Figure 1: **Inference and reconstruction** on test images. For each image, from left to right: input, reconstruction, summary of inferred locations, sprites generated from each appearance latent variable.

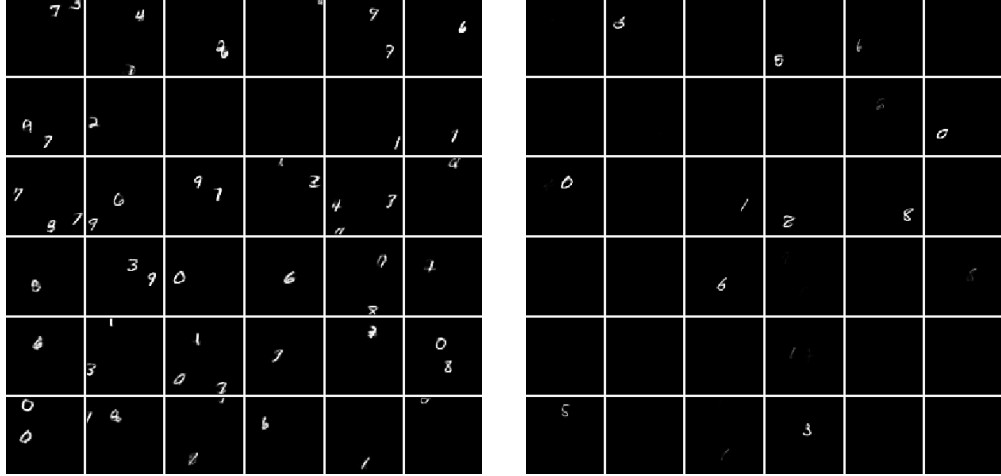

Figure 2: **Generated samples.** Left: images generated by LAVAE from its prior $p(\mathbf{z}_{\mathrm{loc}} \,|\, n)p(\mathbf{z}_{\mathrm{app}} \,|\, n)p(n)$, where $p(n)$ is learned from data. Right: images generated by the baseline from its prior $p(\mathbf{z})$.

can be independently manipulated in the latent space, the qualitative disentanglement experiments in Figure 3 prove that objects are disentangled from one another, and so are the appearance and location of each object.

## 3.2 MULTI-DSPRITES

The multi-dSprites data set is generated similarly to the multi-MNIST ones, by scattering a number of sprites on a black canvas of size $64 \times 64$. Here, the sprites are simple shapes in different colors, sizes, and orientations, as in the dSprites data set (Higgins et al., 2017; Matthey et al., 2017). The shape of each sprite is randomly chosen among square, ellipse, and triangle. The maximum sprite size is $19 \times 19$, and each sprite's scale is randomly chosen among 6 linearly spaced values in $[0.6, 1.0]$. The orientation angle of each sprite is uniformly chosen among 40 linearly spaced values in $[0, 2\pi)$. The color is uniformly chosen among the 7 colors such that the RGB values are saturated (either 0 or 255), and at least one of them is not 0. This means that each color component of each pixel is a binary random variable and can be modelled independently as in the multi-MNIST case (leading to $3D$ terms in the likelihood, instead of $D$).

Figure 4 shows images generated by sampling from the prior in LAVAE and in the VAE baseline, where we can appreciate how our model accurately captures the number and diversity of objects in the data set. Figure 5 shows instead an example of inference on test images, as explained in Section 3.1. Finally, as we did for multi-MNIST, we performed disentanglement experiments where, starting from the latent representation of a test image, we manipulate the latent variables. In Figure 6 we change the order of the location variables to swap objects, whereas Figure 7 shows the effect of separately altering the location and appearance latent variables of a single object.

Table 1: Quantitative results on multi-MNIST and multi-dSprites *test sets*. The log likelihood lower bound is estimated with 100 importance samples. Note that the MNIST-10k dataset is a more complex task than MNIST-1k because the model has to capture a larger variation in appearance. The object count accuracy is measured as the percentage of images for which the inferred number of objects matches the ground truth (which is not available during training).

|  | multi-MNIST-1k | | multi-MNIST-10k | | multi-dSprites | |
|---|---|---|---|---|---|---|
|  | $-\log p(\mathbf{x}) \leq$ | $n$ acc. | $-\log p(\mathbf{x}) \leq$ | $n$ acc. | $-\log p(\mathbf{x}) \leq$ | $n$ acc. |
| LAVAE | $\mathbf{42.2} \pm 1.1$ | $\geq 99.9\%$ | $\mathbf{60.4} \pm 2.8$ | $\geq 99.7\%$ | $\mathbf{43.9} \pm 1.0$ | $\geq 99.9\%$ |
| baseline | $51.6 \pm 0.6$ | — | $64.6 \pm 0.8$ | — | $45.8 \pm 0.7$ | — |

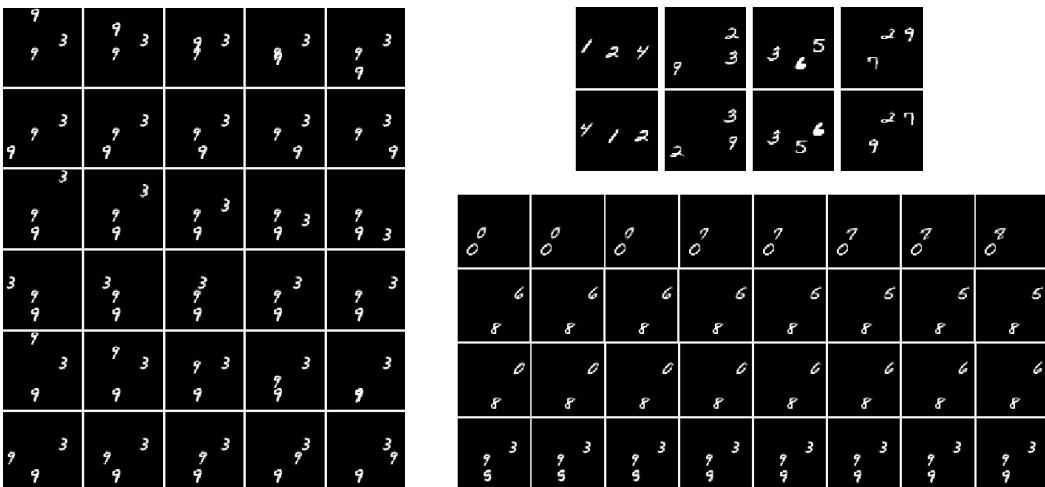

Figure 3: **Disentanglement experiments** on test images. Objects are represented independently of each other, and their location and appearance are disentangled by design. **Left:** Latent traversal on one of the 7 location variables. **Top right:** Reordering the sequence $\{\mathbf{z}_{\mathrm{loc}}^{(i)}\}_i$ or equivalently of $\{\mathbf{z}_{\mathrm{app}}^{(i)}\}_i$ leads to objects being swapped (top row: original reconstruction; bottom row: swapped objects). **Bottom right:** In each row, latent traversal on one of the appearance variables along one dimension.

### 3.3 GENERALIZING TO MORE OBJECTS

As mentioned above, LAVAE correctly infers the number of objects in images from the 7-object versions of our data sets, despite the fact that it was only trained on images with up to 3 objects. Furthermore, by representing each object independently and disentangling location and appearance of each object, it accurately decomposes 7-object scenes and allows intervention as easily as in images with fewer objects. Figure 8 demonstrates this on the 7-object version of the multi-MNIST-10k data set. Finally, in Figure 9 we show images generated by LAVAE after modifying the prior $p(n)$ to be uniform in $\{4, 5\}$.

## 4 RELATED WORK

Our work builds on recent advances in probabilistic generative modelling, in particular variational autoencoders (VAEs) (Kingma and Welling, 2013; Rezende et al., 2014). One of the methods closest to our work in spirit is Attend Infer Repeat (AIR) (Eslami et al., 2016), which performs explicit object-wise inference through a recurrent network that iteratively attends to one object at a time. A limitation of this approach however is that it has not been shown to generalize well to a larger number of objects. Closely related to our work is also the multi-entity VAE (Nash et al., 2017),

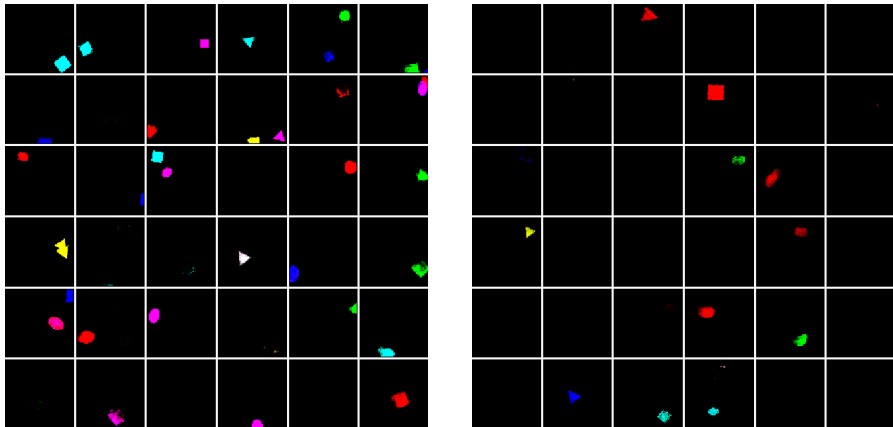

Figure 4: **Generated samples.** Left: images generated by LAVAE from its prior $p(\mathbf{z}_{\text{loc}} \mid n)p(\mathbf{z}_{\text{app}} \mid n)p(n)$, where $p(n)$ is learned from data. Right: images generated by the baseline from its prior $p(\mathbf{z})$.

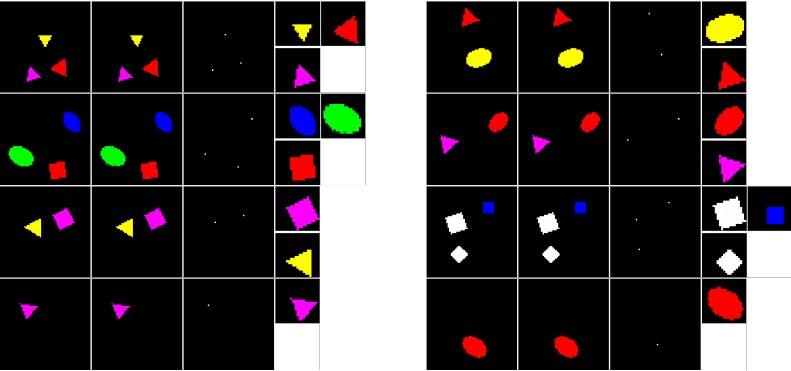

Figure 5: Example of **inference and reconstruction** on multi-dSprites test images. From left to right: input, reconstruction, summary of inferred locations, sprites generated from the inferred appearance latent variables.

in which multiple objects are independently modelled by different latent variables. The inference process does not include an explicit attention mechanism, and uses instead a spatial KL map as a proxy for object presence. Each object's latent is decoded into a full image, and these are aggregated by an element-wise operation, thus the representation of each object's location and appearance are entangled. In the same spirit, the recently proposed generative models MONet (Burgess et al., 2019)

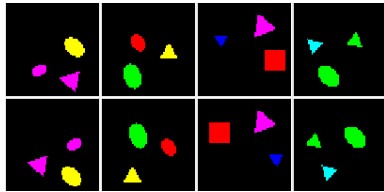

Figure 6: **Object swap.** Changing the order of location (or equivalently appearance) latent variables leads to objects being swapped. Top row: original reconstruction of test image; bottom row: objects are swapped by manipulating the latent variables.

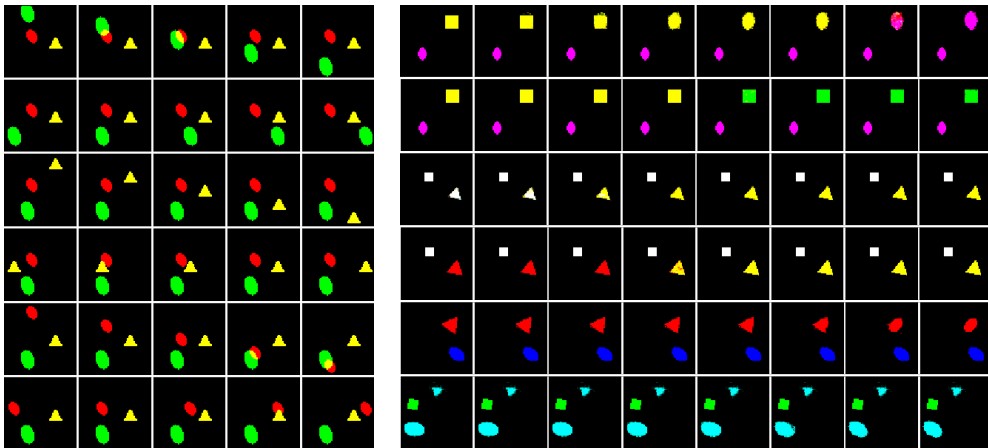

Figure 7: Left: **Location traversal.** Latent traversal on location variables in a test image with 3 objects. Right: **Appearance traversal.** Latent traversal on appearance variables: changing $\mathbf{z}_{\text{app}}^{(i)}$ for some $i$ along one latent dimension corresponds to changing appearance attributes of one specific object. The appearance latent space is only partially disentangled. Here we show examples where a change in one latent dimension leads to a change of a single factor of variation (rows 2, 4, 5, 6). However, in row 1 there is a change both in color and shape, and in row 3 both in color and scale.

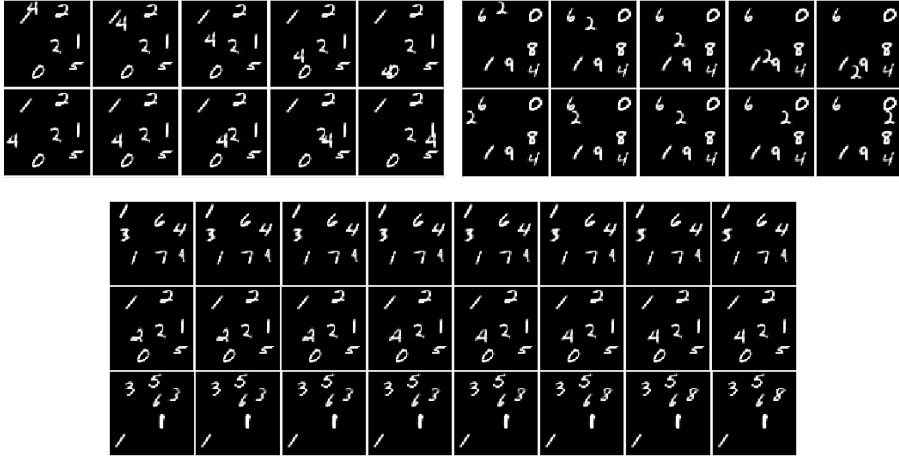

Figure 8: **Disentanglement with more objects.** Latent traversal on location (top) and appearance (bottom) variables, on multi-MNIST-10k test images containing 7 objects. LAVAE can still correctly infer the scene's structure and reconstruct it, allowing intervention on location or appearance of single objects.

and IODINE (Greff et al., 2019) learn without supervision to segment the scene into independent and interpretable object-based parts. Although these are more flexible than AIR and can model more complex scenes, the representations they learn of object location and appearance are not disentangled. All methods cited here are likelihood based so they can and should be compared in terms of test log likelihood. We leave this for future work.

Other unsupervised approaches to visual scene decomposition include Neural Expectation Maximization (Greff et al., 2017; Van Steenkiste et al., 2018), which amortizes the classic EM for a spatial mixture model, and Generative Query Networks (Eslami et al., 2018), that learn representations of rich 3D scenes but do not factor them into objects and need point-of-view information during training. Methods following the vision-as-inverse-graphics paradigm (Poggio et al., 1985; Yuille and Kersten, 2006) learn structured, object-centered representations by making strong assumptions on the latent codes or by exploiting the true generative model (Kulkarni et al., 2015; Wu et al., 2017; Tian et al., 2019). Non-probabilistic approaches to scene understanding include adversarially trained

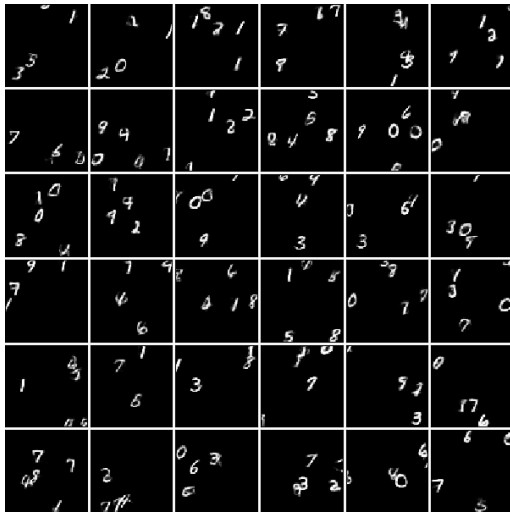

Figure 9: **Generation with fixed number of objects.** Images generated by LAVAE from a modified prior in which $n$ takes value 5 or 6 with probability $1/2$.

generative models (Pathak et al., 2016) and self-supervised methods (Doersch et al., 2015; Vondrick et al., 2018). These, however, do not explicitly tackle representation learning, and often have to rely on heuristics such as region masks. Finally, examples of supervised approaches are semantic and instance segmentation (Ronneberger et al., 2015; He et al., 2017; Jégou et al., 2017; Liu et al., 2018), where acquiring labels for training is typically expensive, and the focus is not on learning structured representations.

## 5 CONCLUSION

We presented LAVAE, a probabilistic generative model for unsupervised learning of structured, compositional, object-based representations of visual scenes. We follow the amortized stochastic variational inference framework, and approximate the latent posteriors by inference networks that are trained end-to-end with the generative model. On multi-MNIST and multi-dSprites data sets, LAVAE learns without supervision to correctly count and locate all objects in a scene. Thanks to the structure of the generative model, objects are represented independently of each other, and the location and appearance of each object are completely disentangled. We demonstrate this in qualitative experiments, where we manipulate location or appearance of single objects independently in the latent space. Our model naturally generalizes to visual scenes with many more objects than encountered during training.

These properties make LAVAE robust to scene complexity, opening up possibilities for leveraging the learned representations for downstream tasks and reinforcement learning agents. However, in order to smoothly transfer to scenes with semantically different components, the appearance latent space should be disentangled. Since in this work we focused on robust model based disentanglement of location and appearance, more work should be done to fully assess and improve disentanglement properties in the appearance model. Another natural extension to this work is to apply it to complex natural images and 3d scenes.

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

## A  KL DIVERGENCE

The KL in Eq. equation 6 can be expanded as follows:

$$
D_{\mathrm{KL}}\big(q(\mathbf{z}_{\mathrm{loc}}, \mathbf{z}_{\mathrm{app}}, n \,|\, \mathbf{x}) \,\big\|\, p(\mathbf{z}_{\mathrm{loc}}, \mathbf{z}_{\mathrm{app}}, n)\big) = \mathbb{E}_q\left[\log \frac{q(\mathbf{z}_{\mathrm{app}} \,|\, \mathbf{z}_{\mathrm{loc}}, n, \mathbf{x})q(\mathbf{z}_{\mathrm{loc}} \,|\, n, \mathbf{x})q(n \,|\, \mathbf{x})}{p(\mathbf{z}_{\mathrm{app}} \,|\, n)p(\mathbf{z}_{\mathrm{loc}} \,|\, n)p(n)}\right]
$$

$$
= \mathbb{E}_{q(\mathbf{z}_{\mathrm{loc}}, \mathbf{z}_{\mathrm{app}}, n \,|\, \mathbf{x})}\left[\log \frac{q(\mathbf{z}_{\mathrm{app}} \,|\, \mathbf{z}_{\mathrm{loc}}, n, \mathbf{x})}{p(\mathbf{z}_{\mathrm{app}} \,|\, n)}\right] + \mathbb{E}_{q(n \,|\, \mathbf{x})}\left[\log \frac{q(n \,|\, \mathbf{x})}{p(n)}\right]
$$

$$
+ \mathbb{E}_{q(\mathbf{z}_{\mathrm{loc}}, n \,|\, \mathbf{x})}\left[\log \frac{q(\mathbf{z}_{\mathrm{loc}} \,|\, n, \mathbf{x})}{p(\mathbf{z}_{\mathrm{loc}} \,|\, n)}\right]
$$

$$
= \mathbb{E}_{q(\mathbf{z}_{\mathrm{loc}} \,|\, n, \mathbf{x})q(n \,|\, \mathbf{x})}\left[\sum_{i=1}^{n} D_{\mathrm{KL}}\Big(q(\mathbf{z}_{\mathrm{app}}^{(i)} \,|\, \mathbf{z}_{\mathrm{loc}}^{(i)}, \mathbf{x}) \,\big\|\, p(\mathbf{z}_{\mathrm{app}}^{(i)})\Big)\right] + D_{\mathrm{KL}}\big(q(n \,|\, \mathbf{x}) \,\big\|\, p(n)\big)
$$

$$
+ \mathbb{E}_{q(n \,|\, \mathbf{x})q(\mathbf{z}_{\mathrm{loc}} \,|\, n, \mathbf{x})}\left[\sum_{i=1}^{n}\Big(\log q\Big(\mathbf{z}_{\mathrm{loc}}^{(i)} \,\big|\, \{\mathbf{z}_{\mathrm{loc}}^{(j)}\}_{j<i}, \mathbf{x}\Big) - \log p\Big(\mathbf{z}_{\mathrm{loc}}^{(i)} \,\big|\, \{\mathbf{z}_{\mathrm{loc}}^{(j)}\}_{j<i}\Big)\Big)\right]
$$

where all expectations can be estimated by Monte Carlo sampling.

## B  IMPLEMENTATION DETAILS

The input image $\mathbf{x}$ is fed into a residual network with 4 blocks having 2 convolutional layers each. Every convolution is followed by a Leaky ReLU and batch normalization. A final convolutional layer outputs the feature map $\mathbf{h}_{\mathrm{app}}$ with $2L$ channels. The output of an identical but independent residual network is fed into a 3-layer convolutional network with one output channel that represents the location logits $\mathbf{h}_{\mathrm{loc}}$. The logits are multiplied by a constant factor $\gamma = 4 \approx 1/2 \log D$. See below for more details about the rescaling of logits.

The number of objects is then inferred by a deterministic function $f_{\mathrm{count}}$ of the location logits. This function filters out points in $\mathbf{h}_{\mathrm{loc}}$ that are not local maxima, then counts the number of points above the threshold $4\gamma$. If $\hat{n}$ is the inferred number of objects in the scene, $f_{\mathrm{count}}$ outputs the corresponding one-hot vector, and the distribution $q(n \,|\, \mathbf{x}) = \delta(n - \hat{n})$ deterministically takes the value $\hat{n}$. We fix the maximum number of objects per image to $M = 10$.

The locations $\{\mathbf{z}_{\mathrm{loc}}^{(i)}\}_{i=1}^{n}$ are iteratively sampled from a categorical distribution with logits $\mathbf{h}_{\mathrm{loc}}$, where all the logits of the previously sampled location are set to a low value to prevent them from being sampled again. At the $i$th step, when the $i$th object's location is sampled, the corresponding feature vector in $\mathbf{h}_{\mathrm{app}}$ is interpreted as the means and log variances of the $L$ components of the $i$th appearance variable. When sampling from the categorical distribution, we use Gumbel-Softmax for single-sample gradient estimation. The temperature parameter $\tau$ is exponentially annealed from 0.5 to 0.02 in 100k steps. The latent space for each appearance variable has $L = 32$ dimensions.

The sprite-generating function $f_{\mathrm{sprite}}$ is a convolutional network that takes as input a sampled appearance vector $\mathbf{z}_{\mathrm{app}}^{(i)}$. The first part of the network consists of a fully connected layer, a convolutional layer, and a bilinear interpolation. The vector $\mathbf{z}_{\mathrm{app}}^{(i)}$ is then expanded and concatenated to the resulting tensor along the channel dimension. The second part of the network consists of 3 residual blocks with 2 convolutional layers each, followed by a final convolutional layer with a sigmoid activation. Leaky ReLU and batch normalization are used after each layer, except in the last residual block. The size of a generated sprite for the multi-MNIST data set is $17 \times 17$, for multi-dSprites it is $21 \times 21$. The mean of the Bernoulli output is then computed from these sprites as explained above.

We optimized the model with stochastic gradient descent, using Adamax with batch size 64. The initial learning rate was 1e-4 or 5e-4 for the location inference network (for the multi-MNIST and multi-dSprites data sets, respectively) and 1e-3 for the rest of the model. In the location inference net, the learning rate was exponentially decayed by a factor of 100 in 100k steps. The model parameters are about 500k, split almost evenly among location inference, appearance inference, and sprite generation.

**Training warm-up.** In practice we found it beneficial to include a *warm-up period* at the beginning of training, in which an auxiliary loss is added to the negative ELBO. We train a fully-convolutional VAE in parallel with our model, and take the location-wise KL in the latent space as a rough proxy for object location, as suggested by Nash et al. (2017). The additional loss is the squared distance between $\mathbf{h}_{\mathrm{loc}}$ and the KL map. The network inferring object location is therefore initially encouraged to mimic a function that is a (rather rough) approximation of the location, and then it is fine-tuned. Intuitively, we are biasing sampling of $\mathbf{z}_{\mathrm{loc}}$ in favor of information-rich locations, which significantly speeds up and stabilizes training. To stabilize training, we also found it beneficial to randomly force $n = 1$ during training with a probability that is 1 during warm-up (30k steps) and then linearly decays to 0.1 in the following 30k steps.

The auxiliary VAE is implemented as a fully convolutional network, in which the encoder consists of 3 residual blocks with downsampling between blocks, and a final convolutional layer. The decoder loosely mirrors the encoder. The latent variables are arranged as a 3D tensor with size $11 \times 11 \times 8$. The auxiliary loss is added to the original training loss for a warm-up phase of 30k steps. In the subsequent 30k steps, the contribution of this term to the overall loss is linearly annealed to 0.

**Rescaling location logits.** Assume we have $k$ objects and the logits of the inferred location distribution are either 0 or $h$. The probability of one of the "hot" locations after softmax is $e^h/(D - k + ke^h) = t$ where $t$ is a constant that should not depend on $D$, and should be close to $1/k$. Solving for $h$ we get $h = \log t + \log(D - k) - \log(1 - kt)$. If $D$ is large enough and $k$ is small enough, we have $\log(D - k) \approx \log D$, and the constant $\log t$ is relatively small in magnitude. Because of our assumptions on the logits and on $t$, we can write $\log(1 - kt) \approx 0$, and therefore the high logits $h$ should be approximately proportional to $\log D$. Thus, using the same fully convolutional network architecture for multiple image sizes, the logits should be scaled by a factor proportional to $\log D$. The threshold for counting objects should likewise follow this rule of thumb.

**Baseline implementation.** As baseline we use a fully convolutional $\beta$-VAE, where the latent variables are organized as a 3D tensor. Being closer in spirit to our method, this leads to a fairer comparison. Furthermore, this inductive bias makes it easier to preserve spatial information. Indeed, this empirically leads to better likelihood and generated samples than a VAE with fully connected layers (and a 1D structure for latent variables), and allows to model more naturally a varying number of objects in a scene. The encoder consists of a convolutional layer with stride 2, followed by 4 residual blocks with 2 convolutional layers each. Between the first two pairs of blocks, a convolutional layer with stride 2 reduces dimensionality. The resulting tensor has size $64 \times 8 \times 8$, and a final convolutional layer brings the number of channels to $2d_{\mathrm{VAE}}/8^2$ where $d_{\mathrm{VAE}}$ is the number of latent variables. The decoder architecture takes as input a sample $\mathbf{z}$ of size $d_{\mathrm{VAE}}/8^2 \times 8 \times 8$ and outputs the pixel-wise Bernoulli means. Its architecture loosely mirrors the encoder's, with convolutions being replaced by transposed convolutions, except for the last upsampling operation which consists of bilinear interpolation and ordinary convolution. All convolutions and transposed convolutions, except for the ones between two residual blocks, are followed by Leaky ReLU and batch normalization. The last convolutional layer is only followed by a sigmoid nonlinearity. In our experiments we used $d_{\mathrm{VAE}} = 1024$ and we linearly annealed $\beta$ from 0 to 1 in 100k steps. The number of model parameters is about 1M.

## C  RESULTS ON MULTI-MNIST-1K

Here we present additional visual results on the multi-MNIST-1k data set, similar to those discussed in the main text.

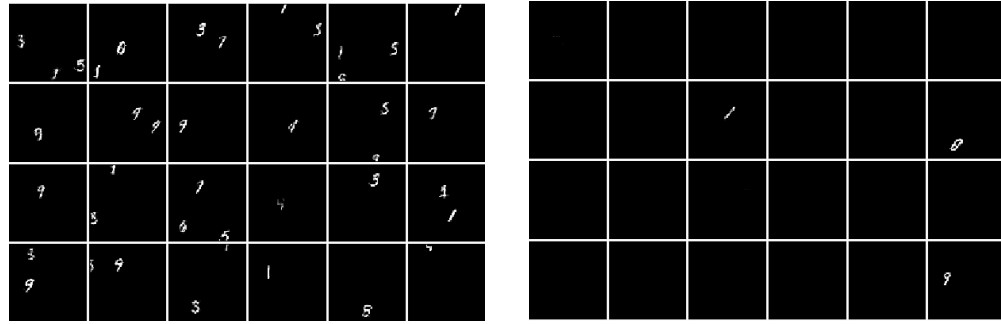

Figure 10: **Prior samples.** Left: images generated by sampling from LAVAE's prior $p(\mathbf{z}_{\mathrm{loc}} \mid n)p(\mathbf{z}_{\mathrm{app}} \mid n)p(n)$, where $p(n)$ is learned from data. Right: images generated by sampling from the baseline's prior $p(\mathbf{z})$.

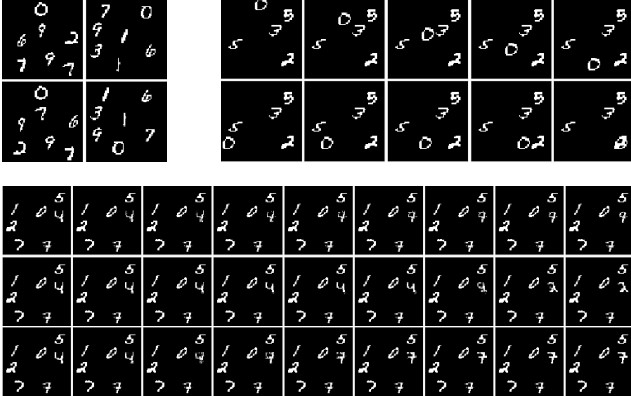

Figure 11: **Disentanglement experiments** on test images with more objects than in the training regime. Objects are represented independently of each other, and their location and appearance are disentangled by design. **Top left:** Reordering the sequence $\{\mathbf{z}_{\mathrm{loc}}^{(i)}\}_i$ or equivalently of $\{\mathbf{z}_{\mathrm{app}}^{(i)}\}_i$ leads to objects being swapped (top row: original reconstruction; bottom row: swapped objects). **Top right:** Latent traversal on one of the 7 location variables. **Bottom:** Latent traversal on one of the 7 appearance variables (along 3 different dimensions).

