# OpenReview forum: "LAVAE: Disentangling Location and Appearance"
_ICLR.cc/2020/Conference — Reject_

### Official Review · AnonReviewer1 · 2019-10-22
**Official Blind Review #1**

**Rating:** 1

**Review:**

This paper introduces a compositional generative model of images, where the image is described by a variable number of latent variables. Moreover, the latent variables are disentangled, in the sense that they represent different parts of the scene, and where appearance and location are described separately. While the generative model is very similar to AIR [1], the central claim of the paper is that the proposed inference scheme can generalize to a much higher number of objects than seen during training, which is demonstrated empirically, and with which AIR struggles. Better generalization is achieved by removing the recurrent core of AIR and replacing it with a fully-convolutional inference network, which predicts appearance vectors at every location in the feature map. The appearance vectors are then stochastically sampled without replacement according to a location distribution. Unlike AIR, this approach does not model the object scale.

I recommend REJECTing this paper. While the improved generalization performance is a useful property, it has been achieved previously and in a very similar fashion by SPAIR [2], which follows a similar model design. SPAIR still uses an RNN, while the proposed approach does not, but this is only a small simplification and does not warrant publication at a top tier conference. There are no other contributions in this paper. Additionally, on the one hand, the experimental evaluation is insufficient: the proposed approach is compared only against a fully-convolutional VAE, while very similar models like AIR [1], SPAIR [2], SuPAIR [3] are not considered. On the other hand, the experimental section focuses on the disentanglement of representations, which is a) evident from the model construction and b) achieved in all previous models.

The paper is clearly written, and the presented generative model is interesting. Having said that, the problem that this paper addresses is mostly solved in [2] and [4]; also both [2] and [4] scale the general approach introduced in [1] to much higher number of objects (in the hundreds) and more general settings (real images, atari games).

[1] Eslami et. al., “Attend, Infer, Repeat:...”, NIPS 2016.
[2] Crawford and Pineau, “Spatially Invariant Unsupervised Object Detection with Convolutional Neural Networks”, AAAI 2019.
[3] Stelzner et. al., “Faster Attend-Infer-Repeat with Tractable Probabilistic Models”, ICML 2019.
[4] Jiang et. al., “Scalable Object-Oriented Sequential Generative Models”, arXiv 2019.

**Experience Assessment:**

I have published in this field for several years.

**Review Assessment: Checking Correctness Of Derivations And Theory:**

I carefully checked the derivations and theory.

**Review Assessment: Checking Correctness Of Experiments:**

I carefully checked the experiments.

**Review Assessment: Thoroughness In Paper Reading:**

I read the paper thoroughly.

---

### Official Review · AnonReviewer3 · 2019-10-22
**Official Blind Review #3**

**Rating:** 3

**Review:**

This paper presents a probabilistic generative model for identifying the location, type and colour of images placed within a grid. The paper is in general well written and presents a large number of visual images demonstrating that the approach works. The main concerns with the paper are as follows:

1) The implementation details for the work are relegated to an appendix. As this is a core concept for the work one would expect this to be presented in the main body of the work.

2) Although there are a large number of visual images there is little in the way of analytical analysis of the work. As a particular concern the authors claim that figure 3 ‘prove’ that objects are disentangled. From Maths 101 I remember it being drummed into us that ‘proof by example is not a proof’.

3) The are parts of the process which are not explained for example, why is the following conducted? ‘The digits are first rescaled from their original size (28 × 28) to 15 × 15 by bilinear interpolation, and finally binarized by rounding.’

4) I would also like to know why this is called disentangling as the whole process prevents the ‘icons’ from overlapping. Disentangle normally refers to things which are at least overlapping. There are examples of works in this area.

5) The example cases are simple. One would expect at least one real-world example.

**Experience Assessment:**

I have read many papers in this area.

**Review Assessment: Checking Correctness Of Derivations And Theory:**

I assessed the sensibility of the derivations and theory.

**Review Assessment: Checking Correctness Of Experiments:**

I assessed the sensibility of the experiments.

**Review Assessment: Thoroughness In Paper Reading:**

I read the paper thoroughly.

---

### Official Review · AnonReviewer2 · 2019-10-24
**Official Blind Review #2**

**Rating:** 1

**Review:**


After reading reviews and comments I have decided to confirm the initial rating.

===================

The work presents an approach to encode latent representations of objects such that there are separate and disentangled representations for location and appearance of objects in a scene.  The work presents impressive qualitative results which shows the practical use of the proposal on multi-mnist and multi-dSprites.

While the use of inference networks proposing positions for the network as a means of improving the disentanglement is clever and seems novel, though not unlike inference sub-networks which are well-known in conditional generation, the evaluation is not up to a standard I can endorse, resulting in a recommendation to reject.

Despite the interesting qualitative results, I will have to quote the work in saying, “All methods cited here are likelihood based so they can and should be compared in terms of test log likelihood. We leave this for future work.”.  Indeed the cited works should have been evaluated against, especially Greff et al. 2019, Nash et al, 2017, and Eslami et al, 2016, which are all very similar.  As written it’s impossible to tell whether this work actually improves over the state of the art, we only have the constructed baseline (which as a community we all know clearly would not have worked).

A figure showing the relevant submodules of the network architecture and what they do in relation to the overall method would be helpful to understand the pipeline and how the inference network relates to the whole.


**Experience Assessment:**

I have read many papers in this area.

**Review Assessment: Checking Correctness Of Derivations And Theory:**

I carefully checked the derivations and theory.

**Review Assessment: Checking Correctness Of Experiments:**

I carefully checked the experiments.

**Review Assessment: Thoroughness In Paper Reading:**

I read the paper thoroughly.

---

### Author Response · Authors · 2019-11-15
**Response to reviewers**

We thank the reviewers for their thorough and helpful comments. The reviewers agree that our approach is interesting and qualitative results are impressive, but also deem the experimental evaluation insufficient, and suggest that an experimental comparison with related methods is needed. We agree with these general points, and we plan to extend the experimental section and compare our results with methods such as AIR [1], SPAIR [2], and SuPAIR [3] in the future.

However, we would like to make a few remarks:

* SCALOR [4] is a concurrent submission to ICLR, and was uploaded to arxiv after the deadline (October 6).

* The papers cited by the reviewers are not really concerned about the generative aspect of the models, and they do not show generated samples or report log-likelihood scores. Comparison with previous methods is mostly qualitative and not necessarily very thorough. For example, in the AIR paper the ELBO is only shown in plots and not explicitly reported, and estimates of the marginal log-likelihood are not mentioned. The SuPAIR paper reports the ELBO for SuPAIR and AIR, but because of the choice of data likelihood, these scores are not comparable, as also stated by the authors. The performance evaluation of SPAIR is entirely focused on counting accuracy and related downstream tasks, overlooking typical generative modeling metrics.

* Real-world examples are rarely considered as they present a different set of difficulties. SCALOR is tested on crowd tracking from surveillance cameras, but the fact that qualitative results on that data set are good is debatable (predicted frames after only 1-2 time steps).

* "Has been achieved previously and in a very similar fashion by SPAIR": there are in fact different mechanisms involved, and experimental results (which are now missing) should show to what extent SPAIR and our model can solve the problem under consideration.

* MNIST digits are rescaled so that more of them can fit in an image, as done in related works. They are binarized so the data can be modeled as a set of independent Bernoulli variables -- a common approach in generative modeling on simple data sets.

* We use the definition of disentanglement used for example in [5] and [6].



[1] Eslami et al., “Attend, Infer, Repeat:...”, NIPS 2016.
[2] Crawford and Pineau, “Spatially Invariant Unsupervised Object Detection with Convolutional Neural Networks”, AAAI 2019.
[3] Stelzner et al., “Faster Attend-Infer-Repeat with Tractable Probabilistic Models”, ICML 2019.
[4] Jiang et al., “Scalable Object-Oriented Sequential Generative Models”, arXiv 2019.
[5] Higgins et al., "beta-VAE: Learning Basic Visual Concepts with a Constrained Variational Framework", ICLR 2017.
[6] Locatello et al., "Challenging Common Assumptions in the Unsupervised Learning of Disentangled Representations", ICML 2019.

---

> ### Comment · AnonReviewer1 · 2019-11-15
> **I'm keeping my score**
>
> Thanks for the response. I appreciate that you are going to compare your method with other baselines. I cannot take your word for it when evaluating the paper, though, and it has no influence on my score.
>
> Other remarks:
> * I do realize that SCALOR was made available only after the ICLR deadline. I cited it because it solves a very similar problem.
> * Even if the other works do not report marginal likelihoods or ELBOs, I don't think that merely reporting them does constitute a big contribution. Moreover, Sequential AIR of Kosiorek et. al. does report ELBOs and estimates of the log-ikelihood for AIR.

---

### Decision · Program_Chairs · 2019-12-19

**Decision:**

Reject

**Comment:**

This paper presents a VAE approach where the model learns representation while disentangling the location and appearance information. The reviewers found issues with the experimental evaluation of the paper, and have given many useful feedback. None of the reviewers were willing to change their score during the discussion period. with the current score, the paper does not make the cut for ICLR, and I recommend to reject this paper.